# The Impact of Exchange Rate Volatility on Exports in Vietnam: A Bounds Testing Approach

**Vinh Nguyen Thi Thuy * and Duong Trinh Thi Thuy**

Foreign Trade University, Hanoi 100000, Vietnam; trinhthithuyduong.ftu@gmail.com
* Correspondence: vinhntt@ftu.edu.vn; Tel.: +84-24-3775-1278 (ext. 112)

**Abstract:** This paper investigates the impact of exchange rate volatility on exports in Vietnam using quarterly data from the first quarter of 2000 to the fourth quarter of 2014. The paper applies the autoregressive distributed lag (ARDL) bounds testing approach to the analysis of level relationships between effective exchange rate volatility and exports. Using the demand function of exports, the paper also considers the effect of depreciation and foreign income on exports of Vietnam. The results show that exchange rate volatility negatively affects the export volume in the long run, as expected. A depreciation of the domestic currency affects exports negatively in the short run, but positively in the long run, consistent with the J curve effect. Surprisingly, an increase in the real income of a foreign country actually decreases Vietnamese export volume. These findings suggest some policy implications in managing the exchange rate system and promoting exports of Vietnam.

**Keywords:** exchange rate; volatility; exports; ARDL; Vietnam

---

## 1. Introduction

In 2015, the exchange rate became a hot issue for Vietnam's economy with regard to concerns about China's devaluation of the Yuan, the increase of the federal fund rate of Fed, and the US dollar appreciation against many currencies in the world. Due to being pegged to the USD, the Vietnam Dong (VND) became more expensive against many foreign currencies, thus, the competitiveness of Vietnamese goods and the trade balance was affected negatively. The debatable policy question is whether a dong-pegged-to-the-dollar policy over the years remains appropriate. From a corporate perspective, does the stability of the VND against USD support enterprises to avoid risk in international business because the US dollar is the main payment currency, or enterprises may be adversely impacted by the uncertainty of bilateral exchange rates for currencies of different countries around the world? In the period of integrating into the world's economy, Vietnam could be seriously challenged by the increase in such risks, therefore, it is necessary to find a suitable exchange rate arrangement. Under that situation, on 31 December 2015, the State Bank of Vietnam issued Decision No. 2730/QD-NHNN to announce the way to determine the central rate of the VND against the USD, which would be used by financial institutions authorized to trade in foreign currencies. The rate is calculated based on three benchmarks: demand and supply of the Vietnam Dong, the movement of eight currencies of the countries having the largest weights for trading and investment with Vietnam, including the USD, the Euro, the Chinese Yuan, the Japanese Yen, the Singapore Dollar, the South Korean Won, the Thai Baht, and the Taiwan Dollar, and macroeconomic balance.

However, whether the exchange rate stability based on the eight major currencies really brings advantages to international trade or not is still a significant question because of the mixed results of theoretical, as well as empirical, studies on the impact of exchange rate volatility on international trade, although many studies also propose that mitigating exchange rate risk is very important to ensuring that the exports of a country achieve sustained stable growth. Moreover, in Vietnam, economists often

study the impact of exchange rate movement on the trade balance, inflation, and economic growth, while studies concentrating on measuring and assessing the influence of exchange rate volatility are still limited, especially in regards to the macro approach. Therefore, it is worthwhile to investigate the effect of exchange rate volatility on exports in the Vietnam context.

For all of the above reasons, this paper investigates the impact of exchange rate volatility between VND and the basket of eight foreign currencies referred to in the central rate benchmark on exports of the Vietnamese economy using quarterly data from the first quarter of 2000 to the fourth quarter of 2014 and the Autoregressive Distributed Lag (ARDL) method of Pesaran et al. (2001). Pesaran's ARDL method shows having comparatively superior forecasting performance compared to the other techniques based on co-integration (Iqbal and Uddin 2013; Adom and Bekoe 2012). The result shows that export performance will be impacted by exchange rate volatility in the long run. A one percent increase in exchange rate volatility will reduce export volume significantly by about 0.11 percent. However, an appreciation of the domestic currency can adversely affect the competitiveness of Vietnamese exports in the international market in the short run, while the Vietnam Dong's devaluation will have positive impacts and improve exports in the long run. A surprising finding is that real foreign income has a negative impact on export volume of Vietnam in both the long run and the short run. The findings provide some implications for managers, policymakers, and entrepreneurs. The remainder of the paper is organized as follows: Section 2 reviews the theoretical background and econometric techniques for examining the effect of exchange rate volatility on exports. Next, Section 3 describes the model, methodology and relevant data used for quantitative assessment in the case of Vietnam. Then, the estimation results are discussed in Section 4. Finally, Section 5 concludes with a summary of findings and policy recommendations.

## 2. Theoretical Framework and Literature Review

Different theories exist in the literature regarding the impact of exchange rate volatility on exporter behavior. An increase in exchange rate volatility may be associated with either an increase or a decrease in the volume of exports, given plausible alternative assumptions.

Traditionally, it has been argued that exchange rate volatility will have a negative influence on exports. Clark (1973) analyses a very early example, in which a firm produces a homogenous commodity and exports its products entirely to one foreign market. In this basic model, the market is considered as perfectly competitive and imported inputs are not required. The firm receives payments for its exports in foreign currency and hedging possibilities are extremely limited. Owing to adjustment costs, the firm cannot change its output over the planning horizon. The unpredictable variation of the exchange rate, therefore, is solely blame for uncertainty about future export sales as well as future profits in domestic currency. For the sake of maximizing the expected value of utility, which depends on both the expected value and the variance of profits, the risk-averse firm would reduce its exposure to risk in response to higher volatility in the exchange rate. That is, the volume of production, and hence exports would be cut down in this circumstance. This simple model is also developed by a number of authors, for example, Baron (1976b); Hooper and Kohlhagen (1978), indicated the same conclusion that exchange rate volatility has a negative effect on exports.

However, all of those conclusions result from several restrictive assumptions. One obvious criticism of the traditional models is that the exporter's risk exposure is attributed solely to the exchange rate volatility, whereas it may depend on the availability of hedging techniques, diversification possibilities, the existence of imported inputs, and other factors. The rationale of this assumption is that forward exchange markets are just in infancy or even not appear in developing economies. In addition, transaction hedging may prove relatively expensive and challenging for some manufacturing firms with a long time between order and delivery. However, this is not the case with advanced countries, in which such markets are well-developed. For risk-adverse entrepreneurs who can hedge their contracts, a higher exchange rate volatility would not always deter exports, as noted by Ethier (1973) and Baron (1976a). Furthermore, the companies can minimize exchange rate risk in other ways; take

multinational cooperation to be a good case in point. Being involved in a wide range of trade and financial transactions over numerous countries, it would see an abundance of diverse opportunities to offset the movement of a bilateral exchange rate, such as the variability of other exchange rates or interest rates. Relaxing the assumption of no imported intermediate inputs, Clark (1973) finds that the loss from the depreciation in a foreign currency to the exporter will be partly alleviated by lowering input cost. Likewise, if inventories are possible and firms can allocate their sales between abroad and home markets, a declining effect on export earnings will also be compensated. More generally, from a finance perspective, Makin (1978) argues that a diversified firm holding a portfolio of assets and liabilities determined in various currencies will be able to protect itself from exchange rate risks related to exports and imports. Finally, recent studies suggest that exchange rate volatility does not just embody a risk, but profit opportunities. For instance, as examined by Canzoneri et al. (1984), if a firm has ability to alter its factor inputs to benefit from changes in exchange rate without adjustment costs, a higher volatility may create greater probability to make profit. Gros (1987) derives a further version of model with the presence of adjustment costs, in which exporting can be seen as an option depending on capacity, taking advantage of favorable conditions (e.g., high prices) and to minimizing the influence otherwise. The value of the option rises as result of higher variability of the exchange rate, creating a positive effect on exports. Therefore, the effect of volatility remains ambiguous because the dominant direction depends on a case-by-case basis.

In the early models, the negative association between exchange rate volatility and expected export increases is supported in terms of risk aversion. The uncertainty of the exchange rate seems to not affect a risk-neutral firm's decision. Nonetheless, De Grauwe (1988) argues that the assumption of risk-averse agents is not adequate to ensure the direction of this link. What is relevant is the degree of risk aversion. An increase in risk, in general, has both a substitution and an income effect that work in opposite directions (Goldstein and Khan 1985). The substitution effect discourages risk-averse agents to export because it lowers the expected utility representing the attractiveness of the risky activity, while the income effects urges very risk-averse agents to increase their exports to avoid the possibility of a severe decline in the revenues. Taken together, these studies support the notion that even though firms are worse off with an increase in exchange rate risk, their response may be to export more rather than less.

All of the theoretical studies reviewed here support the notion that the net effect of exchange rate volatility on exports is ambiguous, as differing results can arise from plausible alternative assumptions and modelling strategies. Increased exchange rate volatility can have no significant effect on exports, or where significant, no systematic effect in one direction or the other.

Numerous empirical studies have been conducted in many countries and areas around the world to evaluate the impact of exchange rate volatility on exports. Again, the implications of the results of those studies confirm that, although exchange rate volatility has an impact on exports, the effect can be either positive or negative depending on the endowment of each country; whether empirical studies use aggregate data, sectoral data or bilateral data; and the econometric techniques applied.

The empirical literature using aggregate data tends to find weak evidence in favor of a negative impact of exchange rate uncertainty on the trade flows of a country to the rest of the world. Using the Engel-Granger method, Doroodian (1999) approximated volatility with both Autoregressive Integrated Moving Average (ARIMA) and Generalized Autoregressive Conditional Heteroskedasticity (GARCH) techniques to study the exports of India, Malaysia, and South Korea from the second quarter of 1973 to the third quarter of 1996. The results reveal significantly negative effects of exchange rate volatility on exports. Meanwhile, employing the Johansen approach of co-integration and using Autoregressive Conditional Heteroskedasticity (ARCH) method to calculate volatility, Arize and Malindretos (1998) found mixed results for two Pacific-Basin countries: volatility is shown to depress New Zealand exports, while its impact is positive in the case of Australia.

To sum up, the majority of empirical studies indicate that the relationship between a single country's exports and exchange rate volatility is statistically significantly negative in the long run,

especially in developing countries, while others consider that there is the positive relationship in the short run or long run. The basis for empirical model development is mostly based on simple demand functions of exports. Relative prices, income, and volatility are often employed as determinants. There are two major problems facing the applied econometrics in these studies. Firstly, there has not yet been a standard exchange rate volatility proxy (Bahmani-Oskooee and Hegerty 2007). Some measure of variance has dominated this field, but the precise calculation of this measure differs from study to study. Later estimates have involved using the standard deviation of a rate of change or the level of a variable. Kenen and Rodrik (1986) draw attention to the moving standard deviation of the monthly change in the exchange rate, which has the advantage of being stationary. Utilizing newer time-series methods, Engle and Granger (1987) developed Autoregressive Conditional Heteroskedasticity (ARCH) as a measure of volatility in time-series errors, which is a widespread measure of exchange rate volatility in the literature. A broader perspective is adopted by (Pattichis 2003) who develops Generalized Autoregressive Conditional Heteroskedasticity (GARCH), which incorporates moving-average processes. These authors' estimates also have the desirable property of stationarity. Some measures are more popular than others, however, none stands out as the standard volatility proxy (Bahmani-Oskooee and Hegerty 2007). The second problem is the type of method used in estimating the empirical model. While the Ordinary Least Squares (OLS) was commonly used in the early papers, newer and more sophisticated techniques, including time-series and panel data methods, in recent studies have facilitated investigation of the sensitivity of exports to a measure of exchange rate volatility. The main goal of modern time-series analysis is to take into consideration integrating properties of the variables so that spurious results can be avoided. Some popular methods of time-series analysis in recent years are the Engle-Granger method, the Johansen method, and the bounds testing approach.

## 3. Exchange Rate Volatility Measurement

Exchange rate volatility denotes the amount of uncertainty or risk about the size of changes in the exchange rate. If the exchange rate can potentially be spread out over a larger range of values in a short time span, it is termed to have high volatility. If the exchange rate does not fluctuate dramatically, and tends to be steadier, it is termed to have low volatility. Additionally, real and nominal exchange rate volatilities are different for practical purposes. The properties of the method used to estimate volatility have also received lots of attention. Bahmani-Oskooee and Hegerty (2007) emphasizes the fact that a clearly dominant approximation for uncertainty has not yet emerged up to now.

In this paper, the exchange rate volatility is measured by the moving average of the standard deviation of exchange rates, which is typically used by a number of scholars such as Chowdhury (1993); Arize and Malindretos (1998); Kasman and Kasman (2005). This equation is as follows:

$$\text{VOL}_t = \left[ \frac{1}{m} \sum_{i=1}^{m} (\text{ER}_{t+i-1} - \text{ER}_{t+i-2})^2 \right]^{\frac{1}{m}} \tag{1}$$

where m is the number of periods; and t is time and ER refers to the exchange rate index. In our study, m = 2.

Bagella et al. (2006) show advantages of effective exchange rate volatility comparing with bilateral exchange rate volatility and find that this variable performs much better than the bilateral exchange rate volatility measure. An important advantage is that the effective exchange rate reflects more sufficiently the stability of a country which might have low bilateral exchange rate volatility with a leading currency but absorb instability via variability of economic policies of its trade partners. Therefore, we use the nominal effective exchange rate between the VND and a foreign currency basket (NEER) to compute the exchange rate volatility. This selected basket consists of eight foreign currencies used by the SBV to refer to the central exchange rate from the beginning of 2016, including: USD (United States), EUR (EU), CNY (China), THB (Thailand), JPY (Japan), SGD (Singapore) KRW (Korea),

and TWD (Taiwan). This option aims to assess the validity of the new exchange rate policy for export purposes. Following the splicing procedure proposed by Ellis (2001), this index is computed as:

$$NEER_t = NEER_{t-1} \frac{\prod_{j=1}^8 \left( NER_t^j \right)^{\omega_{jt}}}{\prod_{j=1}^8 \left( NER_{t-1}^j \right)^{\omega_{jt}}} \tag{2}$$

where $NEER_t$ is the real effective exchange rate of Vietnam at time t; $NER_t^j$ is the nominal bilateral exchange rate relative to currency of country j, measured as the number of units of the domestic currency per unit of currency of country j and expressed as an index; $\omega_{jt}$ is the weight assigned to the currency of country j at time t, reflecting the contribution of the country j to Vietnam's foreign trade, $\sum_{j=1}^8 \omega_{jt} = 1$.

In Equation (2), the nominal effective exchange rate is calculated as the ratio of geometrically weighted bilateral nominal exchange rates in the current period and in the preceding period, using current weights, spliced onto the preceding level of nominal effective exchange rate. There are two main advantages associated with the use of this approach. Firstly, the weights are allowed to vary over time in order to account for the possibility that some countries may become more important trading partners. Otherwise, if actual trade shares move significantly and this is not taken into consideration, the effective exchange rate would give a misleading picture of the net effect of movements in particular bilateral exchange rates. Secondly, as changing weights are updated, it is important that the exchange rate index should be spliced together with the previous observation. Otherwise, in periods in which the weights change, it would not be clear whether a change in the NEER is reflecting changes in the weights or in the bilateral exchange rates, as we can see from a common calculation: $NEER_t = \prod_{j=1}^8 \left( NER_t^j \right)^{w_{jt}}$. There are some prior studies using this approach, such as Moccero and Winograd (2006); Chinn (2006); Betliy (2002); Dullien (2005).

Data for bilateral exchange rates and trade weights are computed from International Financial Statistics (IFS) and Direction of Trade Statistics (DOTS) of IMF.

Figure 1 describes the volatility of NEER of the Vietnam Dong versus the eight currency basket for the period from the first quarter of 2000 to the fourth quarter of 2014. The degree of this volatility depends on the exchange rate policy and the fluctuation of foreign currencies in the world market. As can be seen from Figure 1, the NEER volatility fluctuated gradually from 2000 to 2007, dramatically increased during the following four years and decreased between 2012 and 2014.

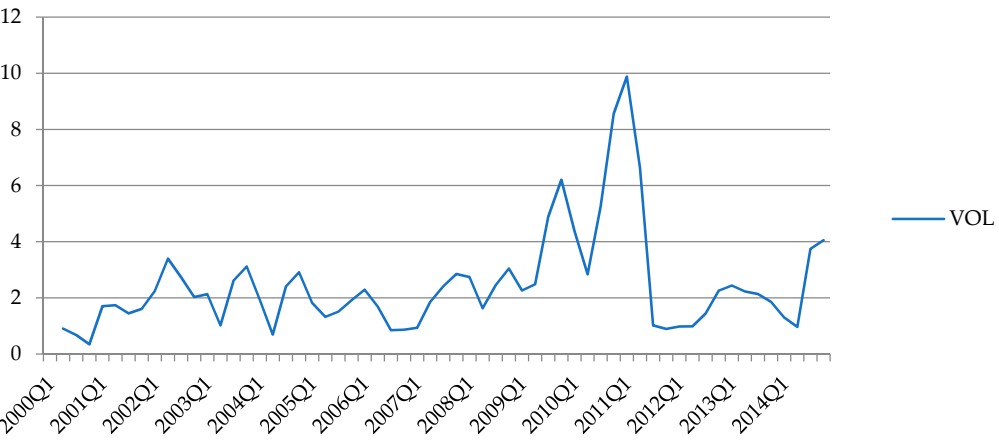

**Figure 1.** Vietnamese NEER volatility (2000 Q1–2014 Q4).

After introducing a new principle for setting the exchange rate in 1999, the volatility from 2000 to 2008 was relatively small as the official exchange rate was almost unchanged. During the period from 2008 to 2011, the State Bank of Vietnam had devalued three times the Vietnam Dong by approximately

10% and adjusted the trading band in commercial banks continuously (widened the band five times from ±0.5% to ±5% and then narrowed it back to 1%). These actions increased the exchange rate fluctuation. From 2012 to 2014, the official exchange rate remained stable except for a devaluation in June 2013 and the trading band was fixed at ±1%, therefore, the volatility was small.

## 4. Empirical Investigation

### 4.1. Econometric Model

According to the microeconomic theory, conventional demand functions are homogenous degree zero in terms of price and income (Deaton and Muellbauer 1980). To examine the impact of exchange rate volatility on exports, this study adds an exchange rate volatility variable to the traditional export demand function comprising consumers' income (or GDP) and relative price, which has been used in many previous studies such as Salas (1982); Gafar (1995); Matsubayashi and Hamori (2003); Ekanayake et al. (2010). The model is specified as follows:

$$X_t = \beta_0 + \beta_1 GDP\_F_t + \beta_2 REER_t + \beta_3 VOL_t + \varepsilon_t \tag{3}$$

where X represents real exports; GDP_F is real foreign income; REER is the real effective exchange rate; and VOL is the exchange rate volatility. With regard to the functional form, Khan and Ross (1977) suggest that a log-linear specification is better than a standard linear one on both empirical and theoretical grounds. That is, the former allows the dependent variable to react proportionally to an increase or decrease in the regressors and exhibits interaction between elasticities. Therefore, all variables in Equation (3) are expressed in logarithmic form. In Equation (3) we have the following expectations for the sign of the regression coefficients: According to the gravity theory of international trade, increases in real GDP of trading partners would be expected to result in greater real exports to those partners, therefore, $\beta_1 > 0$. Due to the relative price effect, the real exchange rate may lead to an increase in the volume of export, therefore, $\beta_2 > 0$. The relationship between the exchange rate volatility and export volume is ambiguous, thus, it is expected that $\beta_3 > 0$ or $\beta_3 < 0$.

When modelling the relationship between a set of time-series variables, it is important to take into account the stationarity of the data. When detecting a spurious regression problem among these series including a unit root, some methods are suggested to solve this problem. One of the simplest ways is taking the differences of the series and estimating a standard regression model. However, this method results in the loss of information that is meaningful for the level relationships. Provided that the first differences of the variables are used, it is impossible to determine a potential long run relationship in levels. Moving from this point, the co-integration approach associated with error-correction modelling was developed during the late 1980s. In this way, both the short run and long run relationship can be analyzed. The co-integration approach developed by Engle and Granger (1987) is suitable for the test based on the expectation of only one co-integrating vector being present. Further, the approach proposed by Johansen (1988) enables researchers to test the case that there is more than one co-integration vector by using the VAR model in which all the variables are accepted as endogenous. However, the important condition that must be met to perform these standard co-integration tests is that all series should not be stationary at levels and they should be integrated of the same order. In order to overcome this problem, Pesaran et al. (2001) have developed the bounds test approach. According to this method, the existence of a co-integration relationship can be investigated between the time-series regardless of whether they are I(0) or I(1) (under the circumstance that the dependent variable is I(1)). This point is the greatest merit of the bounds test over conventional co-integration testing. Moreover, this approach can distinguish dependent and independent variables and is more suitable than another method for dealing with small sample sizes (Ghorbani and Motallebi 2009). In addition, different variables can be assigned different lag lengths as they enter the model.

As reviewed by Bahmani-Oskooee and Hegerty (2007), while common variables in trade models are non-stationary series, most measures of exchange rate volatility are stationary. Therefore, the ARDL approach by Pesaran et al. (2001) is the most highly recommended to investigate the effect of exchange rate volatility on exports. There are some prior studies using this approach, such as De Vita and Abbott (2004); Sekantsi (2008); Yin and Hamori (2011); and Alam and Ahmad (2011). To implement the bounds test procedure, Equation (3) is modelled as a conditional ARDL error correction model as follows:

$$\text{LEX}_t = \alpha_0 + \beta_0 t + \sum_{i=1}^{l_1} \delta_{1i} \text{LEX}_{t-i} \sum_{i=1}^{l_2} \delta_{2i} \text{LGDP\_F}_{t-i} \sum_{i=1}^{l_3} \delta_{3i} \text{LREER}_{t-i} \sum_{i=1}^{l_4} \delta_{4i} \text{LVOL}_{t-i} + u_t \qquad (4)$$

$$\begin{aligned}\Delta\text{LEX}_t = {} & \alpha_1 + \gamma_0 t + \theta_1 \text{LEX}_{t-1} + \theta_2 \text{LGDP}_{Ft-1} + \theta_3 \text{LREER}_{t-1} + \theta_4 \text{LVOL}_{t-1} \\ & + \sum_{i=1}^{l_1-1} \lambda_{1i} \Delta\text{LEX}_{t-i} + \sum_{i=1}^{l_2-1} \lambda_{2i} \Delta\text{LGDP\_F}_{t-i} + \sum_{i=1}^{l_3-1} \lambda_{3i} \Delta\text{LREER}_{t-i} + \sum_{i=1}^{l_4-1} \lambda_{4i} \Delta\text{LVOL}_{t-i} + u_t \end{aligned} \qquad (5)$$

where LEX, LGDP_F, and LREER are the natural logarithms of real exports, real foreign income, and the real effective exchange rate (all data are seasonally adjusted); LVOL is the natural logarithm of the nominal exchange rate volatility of Vietnam; $l_1$, $l_2$, $l_3$, $l_4$ are lag-lengths; $\theta_1$, $\theta_2$, $\theta_3$, $\theta_4$ are long-run coefficients; and $\lambda_{1i}, \lambda_{2i}, \lambda_{3i}, \lambda_{4i}$ are short-run coefficients (if the co-integration vector exists) and $u_t$ is a random disturbance term.

According to Pesaran et al. (2001), the ARDL approach uses two main steps to estimate the level relationship. The first step is the co-integration test to determine whether a level relationship exists between the variables in Equation (4). The null hypothesis of no level relationship among variables is tested. This test is performed on the basis of comparing the computed F-statistic values with bounds on critical values which depend on the number of variables. Furthermore, some later studies propose the critical value table for special cases, such as the study by Narayan (2005) dealing with small sample size. For various situations, those authors give lower and upper bounds on the critical values. In each case, the lower bound is based on the assumption that all the variables are I(0), and the upper bound is based on the assumption that all the variables are I(1). If the estimated F-statistic falls below the lower bound we cannot reject the null hypothesis, so no co-integration is possible. If the F-statistic exceeds the upper bound, we conclude that we have co-integration. Finally, if the F-statistic falls between the bounds, the test is inconclusive. If the long-run relationship is established between the variables, the long-run and short-run coefficients can be obtained by using the ARDL approach. The appropriate lag orders of variables are chosen using the Schwarz Information Criterion (SIC).

*4.2. Data and Data Sources*

The study uses quarterly data from the first quarter of 2000 to the fourth quarter of 2014, including 60 observations. This period is used for research in order to minimize the problems associated with monetary policy changes. Data for real exports (EX) is collected from the General Statistics Office of Vietnam (GSO). Real foreign income (GDP_F) is measured by the export weighted GDP Volume Index of the twenty largest export partners during this period (United States, Japan, China, Australia, Singapore, Germany, Korea, Malaysia, United Kingdom, Philippines, Netherlands, Thailand, Canada, France, Indonesia, Switzerland, Belgium, Hong Kong, Italy, and Spain). GDP_F is calculated as follows:

$$\text{GDP\_F}_t = \prod_{j=1}^{20} (Y_{jt})^{\varphi_{jt}} \qquad (6)$$

where $Y_{jt}$ is the real GDP of each partner, calculated by the GDP Volume Index, collected from the IFS dataset; $\varphi_{jt}$ is the export weight assign to partner j at time t, computed by using data from the DOTS and $\sum_{j=1}^{20} \varphi_{jt} = 1$.

The real effective exchange rate index (REER) is defined in domestic currency terms (an increase in its value indicates a depreciation of Vietnamese currency) and is estimated by the geometric average, as in the following common equation:

$$REER_t = \prod_{j=1}^{n} \left( NER_t^j \frac{CPI_t^j}{CPI_t^{VN}} \right)^{w_{jt}} \tag{7}$$

where $REER_t$ is the real effective exchange rate of Vietnam at time t; n is the number of trading-partner currencies in the trade basket; $NER_t^j$ is the nominal bilateral exchange rate relative to currency of country j, measured as the number of units of the domestic currency per unit of currency of country j and expressed as an index; $CPI_t^j$ and $CPI_t^{VN}$ are consumer price indices at time t of foreign country j and Vietnam, respectively; and $\omega_{jt}$ is the trade-weight assigned to currency of country j at time t, reflecting the contribution of the partner j to Vietnam's foreign trade, $\sum_{j=1}^{n} \omega_{jt} = 1$. Further, with the same rationale as Equation (2), Equation (7) is adjusted according to the splicing procedure proposed by Ellis (2001) to avoid biasing the result due to changing weights.

The currency basket includes the currencies of Vietnam's twenty largest trading partners during the period from 2000 to 2014, which are: USD (United States), JPY (Japan), CNY (China), KRW (Korea), SGD (Singapore), TWD (Taiwan), THB (Thailand), MYR (Malaysia), AUD (Australia), HKD (Hong Kong), IDR (Indonesia), INR (India), GBP (United Kingdom), KHR (Cambodia), PHP (Philippines), RUB (Russia), AED (United Arab Emirates), CHF (Switzerland), CAD (Canada), and EUR (19 Eurozone countries). The basket covered over 90% of Vietnam's total trade in every year since 2000. In addition, each selected partner accounted for at least 0.2 percent of total foreign trade during this period.

Data for trade values is collected from the DOTS, while bilateral exchange rate data and consumer price index data are collected from the IFS of IMF.

### 4.3. Results and Discussions

### 4.3.1. Unit Root Test

Prior to constructing our models, all variables are tested for stationary and the order of integration is determined. Augmented Dicker Fuller (ADF) tests are conducted including a drift term and both with and without a trend. The lag-lengths for ADF regressions are chosen using the Schwarz Information Criterion (SIC). Table 1 contains the results of the unit root tests.

**Table 1.** ADF unit root tests.

| Variable | Level | | First Difference | |
|----------|----------|--------------------|----------|--------------------|
| | **Constant** | **Constant and Trend** | **Constant** | **Constant and Trend** |
| LEX | 0.37 | −1.46 | −9.61 *** | −9.55 *** |
| LGDP_F | −0.64 | −2.98 | −4.92 *** | −4.89 *** |
| LREER | −0.39 | −2.37 | −4.46 *** | −4.61 *** |
| LVOL | −4.19 *** | −4.30 *** | −7.32 *** | −7.32 *** |

Note: *** are respectively significant at 1%.

The results confirm that all the series are I(1), with the exception of the exchange rate volatility (LVOL), which is I(0). In other words, unit root tests show that the dependent variables are I(1) and the independent variables are a mixture of I(0) and I(1). Thus, the ARDL approach is more suitable than other approaches for examining relationships in levels of variables.

### 4.3.2. Bounds Testing for Level Relationships

Using Schwarz (Bayes) criterion to find optimal lags, the EVIEWS 9.5 software developed by IHS Markit (London, UK) suggests the model of ARDL (2,0,2,7).

A key assumption in the Bounds Testing methodology of Pesaran et al. (2001) is that the errors of Equation (6) must be serially independent. This requirement may also be influential in the choice of optimal lags for the variables in the model. We use the LM test to test the null hypothesis of no serial correlation. The result indicates that at the 1% significance level, we cannot reject the null hypothesis, therefore, the selected model is suitable to test the cointegration relationship between the variables.

Due to the small sample size, we use the critical value bounds given by Narayan (2005). The result of bounds testing is shown in Table 2.

**Table 2.** F-statistics to test the existence of long run relationships.

| Model | Number of Regressors | Sample Size | Estimated F Test Value | Critical Values Bounds-Narayan (2005), Unrestricted Intercept and Unrestricted Trend | | | | | |
| --- | --- | --- | --- | --- | --- | --- | --- | --- | --- |
| | | | | 10% | | 5% | | 1% | |
| | k | n | F-Statistic | I(0) | I(1) | I(0) | I(1) | I(0) | I(1) |
| ARDL (2,0,2,7) | 3 | 52 | 14.576 | 3.673 | 4.715 | 4.368 | 5.545 | 5.995 | 7.335 |

As can be seen from the table, the calculated F-statistic of 14.576 exceeds the upper bounds, this supports the existence of level relationships between real exports, real foreign income, real effective exchange rate and exchange rate volatility in the export equation. The selected ARDL model is rewritten as a single error correction model to identify long run and short run relationships.

### 4.3.3. Short run and Long run Relationship

By normalizing the exports, from the ARDL (2,0,2,7), the empirical results of the relationships in levels are presented in Equation (8):

$$\begin{array}{l} LEX_t = 0.047t - 1.403LGDP\_F_t + 0.987LREER_t - 0.102LVOL_t + e_t \\ \quad\quad (0.004) *** \quad\quad (0.344) *** \quad\quad (0.139) *** \quad\quad (0.012) *** \end{array} \tag{8}$$

Note: *** are respectively significant of 1%, the standard errors are in parenthesis.

The estimation result suggests that all of variables could significantly explain the variation in exports at the 1% level of significant.

The estimated coefficient of VOL is about −0.11 percent, implying that the exchange rate volatility has a negative impact on real exports. A one percent increase in the volatility reduces Vietnamese exports by about 0.11%. This is in line with the theoretical models of the behavior of risk adverse exporters in Clark (1973); Kohlhagen (1978), etc.; Arize and Malindretos (1998) argue that higher exchange rate volatility will depress export volume through a rise in adjustment costs like irreversible investment due to higher uncertainty and risks.

At the macro level, this result is consistent with Qian and Varangis (1994) from considering the cases of developing countries. In these countries, the means of payment in international trade is in foreign currency and the degree of dollarization is fairly high (always above 15 percent in Vietnam), hence, the impact of exchange rate volatility is significant to economic activities. In addition, in developing countries like Vietnam, the derivative markets are underdeveloped, so that hedging may not only be limited but also costly. Another possible explanation for the long run negative impact of exchange rate volatility is that the higher the risk, the higher the value of options, leading to increased costs to ensure the future profit. This reduces the transaction volume in the market.

However, for the short run relationships shown in Table 3, all coefficients of the first difference of VOL are positive and statistically significant at the 1% level, indicating that if exchange rate volatility

increases, export volume will increase in the short run. To sum up, the volatility of exchange rate has a positive and significant short-run effect on exports whilst, in the long run, volatility adversely affects export performance in Vietnam. This result is likely to be related to the simple model of De Grauwe (1988) arguing that the effect of an increase in risk can be decomposed into a substitution effect and an income effect. The substitution effect causes risk-averse firms to decrease export activities as the expected marginal utility of export revenues decrease, while the income effect leads risk-averse firms to boost export performance to avoid severe falls in revenues. Kroner and Lastrapes (1993) argue that enterprises may increase commercial activity as they expect the market to deteriorate in the future due to unforeseen fluctuations in the exchange rate. Thus, they quickly trade at the present time, trying to maximize profits to compensate for possible losses. Thus, in the short run, the income effect can offset the substitution effect, so exports will be encouraged. Alternatively, in the long run, enterprises may have more flexible responses to risks, such as transferring export goods to the domestic market and the cost of hedging becomes more expensive, so the substitution effect can dominate the income effect. This results in a decline in exports in the long run.

**Table 3.** The ECM for the selected ARDL model of output equation.

| Regressors | ARDL (2,0,2,7) |
|---|---|
| $\Delta$LEX$(-1)$ | 0.590 (0.116) *** |
| $\Delta$LGDP_F | $-2.653$ (0.312) *** |
| $\Delta$LREER | 1.086 (0.301) *** |
| $\Delta$LREER$(-1)$ | $-0.883$ (0.333) ** |
| $\Delta$LVOL | $-0.001$ (0.001) |
| $\Delta$LVOL$(-1)$ | 0.148 (0.019) *** |
| $\Delta$LVOL$(-2)$ | 0.119 (0.017) *** |
| $\Delta$LVOL$(-3)$ | 0.110 (0.015) *** |
| $\Delta$LVOL$(-4)$ | 0.082 (0.014) *** |
| $\Delta$LVOL$(-5)$ | 0.063 (0.012) *** |
| $\Delta$LVOL$(-6)$ | 0.034 (0.012) *** |
| C | 22.434 (2.225) *** |
| ECM$_{t-1}$ | $-1.709$ (0.172) *** |
| Adj. R-Square | 0.726 |
| $ECM_t = LEX_t - (-1.403 LGDPF_t + 0.987 LREER_t - 0,107 LVOL_t + 0,047t)$ | |

\*\*, \*\*\* are respectively significant at 5% and 1% levels. The t-ratios are in brackets.

Surprisingly, at the 1% level of significance, the coefficient of the real foreign income variable is negative. The estimated result suggests that if real income of the main importing countries from Vietnam goes up by 1%, the export volume of Vietnam will go down by 1.4%. In addition, the GDP_F variable has a negative short-run coefficient, implying that real trading partners' income exerts a significant adverse effect on real exports of Vietnam in both the short run and the long run. This finding is different to the results of previous studies showing that the impact of foreign output on exports in Vietnam is positive. However, this discrepancy may be due to the fact that those studies have used the nominal values of GDP and exports while, in this paper, the data in real terms has been calculated. Moreover, Vietnamese exports remain low-grade in terms of technological content and added value. Most agricultural products and minerals are exported in their raw or preliminarily processed forms, therefore, an increase in the real foreign income may decline the expenditure on Vietnamese goods following the theory of Engel (1857) on necessary goods that the demand decreases as income increases.

The coefficient of the real effective exchange rate is significant at the 1% level in the long run equation, implying if real exchange rate increases by 1%, the export volume will increase by 0.99%. Therefore, an appreciation will hamper export performance in Vietnam. This is in line with the theory and many empirical studies suggesting that the REER value represents the competitiveness of Vietnamese goods in the international market. Nonetheless, the short-run coefficient of the REER

variable is negative and highly significant. Thus, a depreciation of the domestic currency affects exports negatively in the short run, but positively in the long run, consistent with the J curve effect.

Table 3 provides the summary of the error correction representation of the estimated ARDL model. The empirical results indicate that the error correction term has the correct sign (negative) and is statistically significant. This is further evidence of co-integration relationships among the variables in the model. The estimated value of the error correction term implies that the speed of adjustment to the long run equilibrium in response to the disequilibrium caused by short-run shocks of the previous period is 170.5% in the export equation.

### 4.3.4. Diagnostic Testing

The diagnostic tests including the normality test, serial correlation test, and heteroscedasticity test generally provide satisfactory outcomes.

Finally, the stability of the long-run coefficients along with the short run dynamics are evaluated by applying the CUSUM and CUSUMSQ (Brown et al. 1975). The CUSUM test uses the cumulative sum of recursive residuals, whereas the CUSUMSQ test is based on the cumulative sum of the squared recursive residuals. As shown in Figure 2 both plots of CUSUM statistics and CUSUMSQ statistics stay within the critical bounds of the 5% significance level (represented by the pair of straight lines drawn at the 5% level of significance). These tests indicate no evidence of any significant structural instability. Therefore, the estimated results are stable over the studied time period.

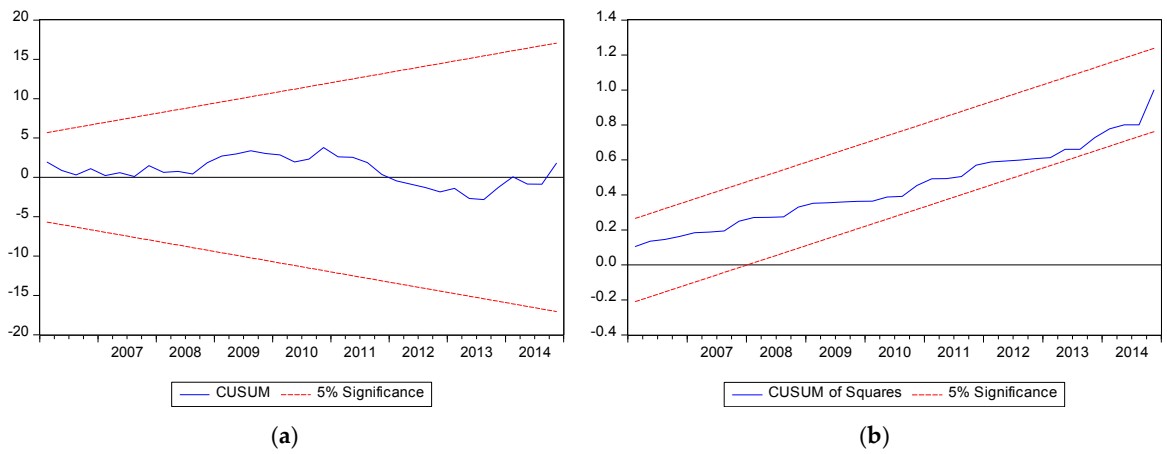

(**a**)　　　　　　　　　　　　　　　　　　　　　　　　(**b**)

**Figure 2.** Plot of cumulative sum of recursive residuals recursive residuals (**a**) and cumulative of squares of recursive residuals (**b**). The straight lines represent critical bounds at the 5% significance level.

## 5. Conclusions

This paper aims to examine the impact of exchange rate volatility on Vietnamese exports performance during the period from the first quarter of 2000 to the fourth quarter of 2014. We use the Moving Average of Standard Deviation (MASD) model and nominal effective exchange rates computed by a weighted average of the nominal bilateral exchange rate of the Vietnam Dong against the basket of eight foreign currencies to measure exchange rate volatility. This paper also applies an approach called the Autoregressive Distributed Lag (ARDL) to investigate the existence of a level relationship among variables in the model and implements it using EVIEWS software offered by IHS Markit (London, UK). The advantage of this approach is that it is suitable for small sample size and regressors which are a mixture of I(0) and I(1).

It is found that there exists a co-integration relationship between real exports, real foreign income, real effective exchange rate, and nominal exchange rate volatility. In addition, the speed of adjustment to the long run equilibrium is fairly high.

The result shows that export performance will be impacted by exchange rate volatility in the long run. A one percent increase in exchange rate volatility will reduce export volume significantly by about 0.11 percent. One anticipated finding is that real foreign income has a negative impact on export volume of Vietnam in both the long run and the short run. As the income of trading partners increases, they tend to import fewer Vietnamese goods, which reflects that the position of Vietnamese goods in the international market remains low-grade. Finally, an appreciation of the domestic currency can adversely affect the competitiveness of Vietnamese exports in the international market in the short run, while the Vietnam Dong's devaluation will have positive impacts and improve exports in the long run. Since the inflation rate of Vietnam is unstable, it may impact the exporters' expectations of movement of real exchange rate. We will deal with this issue for our future empirical research.

These findings have some important policy implications. Firstly, for the State Bank of Vietnam, the conversion of the exchange rate regime from announcing a solid exchange rate between the VND and the US to announcing a central rate and cross rates with eight strong currencies is the right direction to promote export performance. The weighting for these currencies in the basket may be calculated based on stabilizing the nominal effective exchange rate with these eight currencies to reduce exchange rate uncertainty.

Secondly, besides considering exchange rate policy, it is essential for the government to adapt synchronous implementation solutions to overcome the bottlenecks in Vietnamese exports. Production cost, brand value, product quality, and technology content are key factors which threaten to decrease export competitiveness.

Finally, in the context of Vietnam, as the foreign currency derivatives market has not fully developed and there are potential risks in international business, enterprises needs a proper international trade strategy, including a long-term vision for risk analysis and forecasting, combined with the flexible use of risk hedging tools such as futures, options, swap contracts. In addition, exporters wishing to promote their international trade should not rely solely on the devaluation of the domestic currency, but on the long-term strategy in building their brand, defining their comparative advantages and increasing market access.

**Author Contributions:** V.N.T.T. designed the model and the computational framework and analyzed the data. D.T.T.T. carried out the implementation. V.N.T.T. and D.T.T.T. performed the calculations. D.T.T.T. wrote the initial manuscript with input from all authors. V.N.T.T. was in charge of the overall direction and planning, as well as revised the manuscript.

**Funding:** This research received no external funding.

**Acknowledgments:** We are grateful to three anonymous referees for their helpful comments and suggestions.

**Conflicts of Interest:** The authors declare no conflict of interest.

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
