# Peer review of "The Impact of Exchange Rate Volatility on Exports in Vietnam: A Bounds Testing Approach"

_jrfm, doi:10.3390/jrfm12010006_

Reviewer 1 Report

I think that the paper analyzes an important problem. I hope that my comments will be of some help for authors to improve the paper.

It is advisable that authors should refer to related research articles more. Some include the following:

Khan, M. S. and Ross, K. Z. (1977) The functional form of the aggregate import demand equation, Journal of International Economics, Vol.7, pp.149-160.

Yin, F. and Hamori, S. (2011) 'Estimating the import demand function in the autoregressive distributed lag framework:The case of China', Economics Bulletin, Vol. 31 no.2 pp. 1576-1591.

Matsubayashi, Y. and Hamori, S., (2003) Some International Evidence on the Stability of Import Demand Function, Applied Economics, Vol. 35, pp. 1497-1504.

Author Response

Dear The Reviewer;

Thank you so much for spending your precious time to read our paper and to give us useful comments.

We would like to reply your comments as follows.

 “It is advisable that authors should refer to related research articles more. Some include the following:

Khan, M. S. and Ross, K. Z. (1977) The functional form of the aggregate import demand equation, Journal of International Economics, Vol.7, pp.149-160.

Yin, F. and Hamori, S. (2011) 'Estimating the import demand function in the autoregressive distributed lag framework:The case of China', Economics Bulletin, Vol. 31 no.2 pp. 1576-1591.

Matsubayashi, Y. and Hamori, S., (2003) Some International Evidence on the Stability of Import Demand Function, Applied Economics, Vol. 35, pp. 1497-1504.”

Response

We read above papers and cited them in our paper. We cited the paper of Khan and Rose (1977) from the line 322 to 326 of the revised paper to explain why the log-linear form is applied in equation (4) and (5). For the paper of Yin and Hamori (2011), we cited in line 360 to provide more evidence of advantages of the ARDL method. For the paper of Matsubayashi and Hamori (2003), we cited it in line 319 to confirm the rationality of the export function.

Our manuscript checked by a native English speaking professors from the Latrobe University.

Again, thank you so much for your support.

Yours sincerely,

The authors of the paper titled “The impact of exchange rate volatility on exports in Vietnam: A bounds testing approach”

Reviewer 2 Report

The paper investigates the impact of exchange rate volatility on exports in Vietnam using 11 quarterly data from the first quarter of 2000 to the fourth quarter of 2014.

I think the work is interesting and well written The findings also suggest some policy  implications in managing the exchange rate system and promoting exports of Vietnam

 In my opinion the paper is acceptable for publication with some minor revisions.

Comments.

The data set used in the analysis spams the period from 2000 to 2014 (60 observations). Is there a reason for this choice? In my opinion a larger data set should be more interesting.

Is it possible to use the proposed approach in a forecasting context?  In this case, the Authors could at least mention this possibility in their conclusions.

In equation (7) the Authors should  specify  how the weights have been determined. They refer to the spliced Laspeyres index method proposed by Luci (2001) to avoid biasing the result  due to changing weights. In my opinion the Authors should further clarify the  implemented procedure.  Are there any other procedures in the literature? What is the advantage of the proposed procedure?

In my opinion the authors should refer to more recent papers on the topic.

Author Response

Dear The Reviewer;

Thank you so much for spending your precious time to read our paper and to give us useful comments.

We would like to reply your comments as follows.

Point 1: The period from 2000 to 2014 (60 observations). Is there a reason for this choice? the period from 2000 to 2014 (60 observations). Is there a reason for this choice? In my opinion a larger data set should be more interesting.

Response 1

We chose the data since 2000 because there was a change in the Vietnamese exchange rate system in 1999. Before March 1999, the central rate was set daily by central bank. After that the rate was set daily at the average of interbank exchange rates on the previous transaction day. This new arrangement has been viewed by Vietnam authorities as a turning point in exchange rate policy in Vietnam, moving closer to a flexible exchange rate system. Moreover, before the year 2000, the exchange rate was set by the central bank, so it was not suitable for testing economic theories based on market orientation.

Data is used until 2014 because: Firstly, at the time we studied, the availability of data to ensure the calculation of the effective exchange rate indices and the real foreign income is until 2014. Secondly, considering the sequence of data will be reasonable for the periods of 10, 15, 20,... years. Thirdly, since 2016, the exchange rate system of Vietnam has moved to the new system based on a basket of currencies. Therefore, we believe that if updating one year (2015) compared with the 15-year study period used, the research results will not affected. If the data is updated to 2017, the structural change in the exchange rate system will be difficult to handle because two years are not long enough to look at the long-term effects of structural change.

We fully agree with the reviewer on the point that a larger data set will make more interesting. However, for the above reasons, we would like to maintain our data set of 15 years from 2000 to 2014.

Point 2: Is it possible to use the proposed approach in a forecasting context? In this case, the Authors could at least mention this possibility in their conclusions

Response 2

Yes, the Pesaran’s ARDL is useful and good method for forecasting. For example,

Iqbal and Uddin (2013) employed data from a broad set of countries including both developed and developing countries to forecast the monetary aggregate (M2) for short, medium and long run horizons. This paper compared the accuracy of the three alternative error correction models such as Engle-Granger (1987) two step procedure, the Johansen (1988) multivariate system based technique and the ARDL based technique of Pesaran et al. (2001). This paper also compared the forecasting performance of ECM with other well-known univariate and multivariate forecasting techniques which do not impose co-integration restrictions such as the ARIMA and the VAR techniques. The results indicate that, in general, for short run forecasting non co-integration based techniques (i.e. unrestricted VAR and ARIMA) result in superior forecasting performance whereas for long run forecasting ECM based techniques perform better. Among the co-integration based techniques, there is an evidence of comparatively superior forecasting performance of the ARDL based error correction model.

Adom and Bekoe (2012) investigated the factors that affect aggregate electricity demand in Ghana both in the short and long-run as a guide for demand-side management based on two econometric approaches - ARDL and PAM. The paper confirmed the superiority of the ARDL approach for forecasting compared to the Partial adjustment approach.

We have added information and evidence of the superiority of the ARDL approach in the Introduction of the revised paper (see line 53 to 55 of the revised paper)

References

(Adom and Bekoe 2012) Adom, Philip Kofi, and William Bekoe. 2012. Conditional dynamic forecast of electrical energy consumption requirements in Ghana by 2020: a comparison of ARDL and PAM. Energy 44: 367-80.

(Iqbal and Uddin 2013) Iqbal, J, and MN  Uddin. 2013. Forecasting accuracy of error correction models: International evidence for monetary aggregate M2. J. Int. Glob. Econ. Stud 6: 14-32.

Point 3: In equation (7) the Authors should  specify  how the weights have been determined. They refer to the spliced Laspeyres index method proposed by Luci (2001) to avoid biasing the result  due to changing weights. In my opinion the Authors should further clarify the  implemented procedure.  Are there any other procedures in the literature? What is the advantage of the proposed procedure?

Response 3

The real effective exchange rate index (REER) is defined in domestic currency terms (an increase in its value indicates a depreciation of Vietnamese currency). Following the splicing procedure proposed by Ellis (2001), this index is computed as:

where 

 is the real effective exchange rate of Vietnam at time ;  is the number of trading-partner currencies in the trade basket;

is the bilateral real exchange rate relative to currency of country ;

is the bilateral nominal exchange rate relative to currency of country , measured as the number of units of the domestic currency per unit of currency of country  and expressed as an index;

 and  are consumer price indices at time  of foreign country  and Vietnam respectively;  is the trade-weight assigned to currency of country  at time , reflecting the contribution of the country  to Vietnam’s foreign trade,  

In the above equation, the real effective exchange rate is calculated as the ratio of geometrically weighted bilateral real exchange rates in the current period and in the preceding period, using current weights, spliced onto the preceding level of real effective exchange rate.

There are two main advantages associated with the use of this approach. Firstly, the weights are allowed to vary over time in order to account for the possibility that some countries may become more important trading partners. Otherwise, if actual trade shares move significantly and this is not taken into consideration, the effective exchange rate would give a misleading picture of the net effect of movements in particular bilateral exchange rates. Secondly, as changing weights are updated, it is important that the exchange rate index should be spliced together with the previous observation. Otherwise, in periods in which the weights change, it would not be clear whether a change in the  is reflecting changes in the weights or in the bilateral exchange rates, as we can seen from a common calculation:

Therefore, by splicing together the series in the way described in equation (7), our study can avoid biasing the result due to changing weights.

There are some prior studies using this approach, such as Moccero and Winograd (2006), Chinn (2006), Betliy (2002), Dullien (2005).

We specified how the weights have been determined and further clarified the implemented procedure of effective exchange rate calculations in equation (2) and (7). Some previous studies are cited. Please see the added contents from line 279 to 296 of the revised paper.

Reference

(Betliy 2002) Betliy, Oleksandra. 2002. Measurement of the real effective exchange rate and the observed J-curve: case of Ukraine. The National University of “Kyiv-Mohyla Academy.

(Chinn 2006) Chinn, Menzie D. 2006. A primer on real effective exchange rates: determinants, overvaluation, trade flows and competitive devaluation. Open economies review 17: 115-43.

(Dullien 2005) Dullien, Sebastian. 2005. China's changing competitive position: Lessons from a unit-labor-cost-based REER. International Trade.

(Ellis 2001) Ellis, Luci. 2001. Measuring the real exchange rate: Pitfalls and practicalities. Reserve Bank of Australia.

(Moccero and Winograd 2006) Moccero, Diego N, and Carlos Winograd. Year. Real exchange rate volatility and exports: Argentine perspectives. Paper presented at the Fourth Annual Conference of the Euro-Latin Study Network on Integration and Trade (ELSNIT) Paris.

http://citeseerx.ist.psu.edu/viewdoc/download?doi=10.1.1.400.6435&rep=rep1&type=pdf

Point 4: In my opinion the authors should refer to more recent papers on the topic.

Response 4

We cited more recent papers on the topic and added them to list of References.

Again, thank you so much for your support.

Yours sincerely,

The authors of the paper titled “The impact of exchange rate volatility on exports in Vietnam: A bounds testing approach”

Reviewer 3 Report

Why is the sample truncated in 2014?. What is novel about this method? Vietnam had very large spikes in inflation over your sample period. To the extent that exporters are smart they would form expectations of future real exchange rate  based on inflation forecasts  given that it takes time to export. A reduced form may be unstable and misleading.eg the possibly wrong sign on foreign real income. e

Author Response

Dear The Reviewer;

Thank you so much for spending your precious time to read our paper and to give us useful comments.

We would like to reply your comments as follows.

Point 1: Why is the sample truncated in 2014?

Response 1

We chose the data since 2000 because there was a change in the Vietnamese exchange rate system in 1999. Before March 1999, the central rate was set daily by central bank. After that the rate was set daily at the average of interbank exchange rates on the previous transaction day. This new arrangement has been viewed by Vietnam authorities as a turning point in exchange rate policy in Vietnam, moving closer to a flexible exchange rate system. Moreover, before the year 2000, the exchange rate was set by the Central Bank, so it was not suitable for testing economic theories based on market orientation.

Data is used until 2014 because: Firstly, at the time we studied, the availability of data to ensure the calculation of the effective exchange rate indices and the real foreign income is until 2014. Secondly, considering the sequence of data will be reasonable for the periods of 10, 15, 20,... years. Thirdly, since 2016, the exchange rate system of Vietnam has moved to the new system based on a basket of currencies. Therefore, we believe that if updating one year (2015) compared with the 15-year study period used, the research results will not affected. If the data is updated to 2017, the structural change in the exchange rate system will be difficult to handle because two years are not long enough to look at the long-term effects of structural change.

For the above reasons, we would like to maintain data set of 15 year from 2000 to 2014.

Point 2: What is novel about this method?

Response 2

Firstly, the bounds testing approach introduced by Pesaran et al. (2001) is advanced approach and has numerous advantages compared to other co-integration based techniques. According to this method, the existence of a co-integration relationship can be investigated between the time-series regardless of whether they are I(0) or I(1) (under the circumstance that the dependent variable is I(1)). This point is the greatest merit of the bounds test over conventional co-integration testing.

Secondly, this approach can distinguish dependent and independent variables and is more suitable than another method for dealing with small sample size (Ghorbani and Motallebi 2009).

Thirdly, different variables can be assigned different lag lengths as they enter the model.

Fourthly, the Pesaran’s ARDL method is evidenced having comparatively superior forecasting performance compared to the other techniques based on co-integration (Iqbal and Uddin 2013, Adom and Bekoe 2012).

Fifthly, as reviewed by Bahmani-Oskooee and Hegerty (2007), while common variables in trade  models are non-stationary series, most measures of exchange rate volatility are stationary (i.e. I(0)). Therefore, the ARDL approach by Pesaran et al. (2001) is the most highly recommended to investigate the effect of exchange rate volatility on exports

References

(Adom and Bekoe 2012) Adom, Philip Kofi, and William Bekoe. 2012. Conditional dynamic forecast of electrical energy consumption requirements in Ghana by 2020: a comparison of ARDL and PAM. Energy 44: 367-80.

(Bahmani-Oskooee and Hegerty 2007) Bahmani-Oskooee, Mohsen, and Scott. W. Hegerty. 2007. Exchange rate volatility and trade flows: a review article. Journal of Economic studies 34: 211-55.

(Iqbal and Uddin 2013) Iqbal, J, and MN  Uddin. 2013. Forecasting accuracy of error correction models: International evidence for monetary aggregate M2. J. Int. Glob. Econ. Stud 6: 14-32.

(Ghorbani and Motallebi 2009) Ghorbani, Mohammad, and Marzieh  Motallebi. 2009. Application Pesaran and Shin Method for Estimating IransImport Demand Function. Journal of Applied Sciences 9: 1175-79.

Point 3: Vietnam had very large spikes in inflation over your sample period. To the extent that exporters are smart they would form expectations of future real exchange rate  based on inflation forecasts  given that it takes time to export. A reduced form may be unstable and misleading.eg the possibly wrong sign on foreign real income. e

Response 3

Firstly, the inflation rate in Vietnam over our sample period is unstable (Table 1).  In some years, inflation rate is high but in some years inflation rate is low, even negative. Therefore, the exporters seem to be difficult to estimate about future inflation.

Table 1: Inflation rate in Vietnam from 2000 to 2014

Year

2000

2001

2002

2003

2004

2005

2006

2007

2008

2009

2010

2011

2012

2013

2014

Target (%)

6

< 5

3-4

< 5

< 5

< 6,5

< 8

< 8

< 10

< 15

7-8

< 7

< 10

7-8

7

Actual (%)

-0.6

0.8

4

3

9.5

8.4

6.6

12.6

19.9

6.5

11.8

18.1

6.8

6.04

1.84

Source: General Statistics Office of Vietnam

Secondly, the paper tried to find the long run relationship among variables, therefore in the unstable inflation period, inflation expectation does not affect significant to export decisions in long run.

Thirdly, the finding of negative coefficient of the real foreign income variable is suitable with Vietnam situation. Vietnamese exports remain low-grade in terms of technological content and added value. Most agricultural products and minerals are exported in their raw or preliminarily processed forms, therefore, an increase in the real foreign income may decline the expenditure of foreigners on Vietnamese goods following the theory of Engel (1857) on necessary goods that the demand decreases as income increases.

References

(Engel 1857) Engel, Ernst. 1857. Die produktions-und konsumptionsverhältnisse des königreichs sachsen. Zeitschrift des Statistischen Bureaus des Königlich Sächsischen Ministeriums des Innern 8: 1-54.

Again, thank you so much for your support.

Yours sincerely,

The authors of the paper titled “The impact of exchange rate volatility on exports in Vietnam: A bounds testing approach”

Round  2

Reviewer 3 Report

Firstly, the inflation rate in Vietnam over our sample period is unstable (Table 1).  In some years, inflation rate is high but in some years inflation rate is low, even negative. Therefore, the exporters seem to be difficult to estimate about future inflation.

This answer to my economic point does not answer the point raised.  The authors are reluctant to employ instruments to forecast the future real exchange rate which is relevant to their decision. As it stands the equation estimated is actually a data mining excercise employing a well known technique.

Author Response

Dear The Reviewer;

Thank you so much for spending your precious time to read our paper and to give us useful comments.

We would like to reply your comments as follows.

Firstly, the inflation rate in Vietnam over our sample period is unstable (Table 1).  In some years, inflation rate is high but in some years inflation rate is low, even negative. Therefore, the exporters seem to be difficult to estimate about future inflation.

This answer to my economic point does not answer the point raised.  The authors are reluctant to employ instruments to forecast the future real exchange rate which is relevant to their decision. As it stands the equation estimated is actually a data mining excercise employing a well known technique.

The main objective of our paper is to find the impact of volatility of nominal effective exchange rate based on the eight major currencies on export which can provide useful information for exchange rate policy. Because since 2016, State Bank of Vietnam has tried to stabilize the official exchange rate based on the eight major currencies. Real exchange rate is not main variable which our paper focus on.

Moreover, previous studies considering the impact of real exchange rates on exports used bilateral or multilateral rates based on the definition of the exchange rate which is same as the calculation formula we used. Compared with these studies, our study tried to do better by using of currency basket with more trading partners, the trade-weight changed over time, and calculating by procedure proposed by Ellis (2001) to avoid biasing the result due to changing weights.

We found some studies tried to forecast the equilibrium exchange rate to consider the impact of real exchange rate misalignment on exports using BEER or FEER approachs. However, this is not our research question.

Because the inflation rates of Vietnam is unstable and sometimes negative, we will tried to deal with this issue in our future work on real exchange rate. We added this issue in the conclusion of the revised manuscript (from line 547 to 549).

For the above reasons, we would like to maintain the way to calculate REER for this study.

Again, thank you so much for your support.

Wish you a Merry Christmas and Happy New Year 2019.

Yours sincerely,

The authors of the paper titled “The impact of exchange rate volatility on exports in Vietnam: A bounds testing approach”
